# Amphotericin B Polymer Nanoparticles Show Efficacy against *Candida* Species Biofilms

**DOI:** 10.3390/pathogens11010073

**Published:** 2022-01-07

**Authors:** Abdulghani Alakkad, Paul Stapleton, Corinna Schlosser, Sudaxshina Murdan, Uchechukwu Odunze, Andreas Schatzlein, Ijeoma F. Uchegbu

**Affiliations:** 1UCL School of Pharmacy, University College London (UCL), 29-39 Brunswick Square, London WC1N 1AX, UK; a.alakkad@ucl.ac.uk (A.A.); p.stapleton@ucl.ac.uk (P.S.); c.schlosser@ucl.ac.uk (C.S.); s.murdan@ucl.ac.uk (S.M.); uchechukwu.odunze.14@alumni.ucl.ac.uk (U.O.); a.schatzlein@ucl.ac.uk (A.S.); 2Nanomerics Ltd., 6th Floor, 2 London Wall Place, London EC2Y 5AU, UK

**Keywords:** *Candida*, *C. albicans*, *C. glabrata*, biofilm, N-palmitoyl-N-monomethyl-N,N-diemthyl-N,N,N-trimethyl-6-O-glycolchitosan, GCPQ, nanoparticles

## Abstract

Purpose: Chronic infections of *Candida albicans* are characterised by the embedding of budding and entwined filamentous fungal cells into biofilms. The biofilms are refractory to many drugs and *Candida* biofilms are associated with ocular fungal infections. The objective was to test the activity of nanoparticulate amphotericin B (AmB) against *Candida* biofilms. Methods: AmB was encapsulated in the Molecular Envelope Technology (MET, N-palmitoyl-N-monomethyl-N,N-dimethyl-N,N,N-trimethyl-6-O-glycolchitosan) nanoparticles and tested against *Candida* biofilms in vitro. Confocal laser scanning microscopy (CLSM) imaging of MET nanoparticles’ penetration into experimental biofilms was carried out and a MET-AmB eye drop formulation was tested for its stability. Results: MET-AmB formulations demonstrated superior activity towards *C. albicans* biofilms in vitro with the EC50 being ~30 times lower than AmB alone (EC50 MET-AmB = 1.176 μg mL^−1^, EC50 AmB alone = 29.09 μg mL^−1^). A similar superior activity was found for *Candida glabrata* biofilms, where the EC50 was ~10× lower than AmB alone (EC50 MET-AmB = 0.0253 μg mL^−1^, EC50 AmB alone = 0.289 μg mL^−1^). CLSM imaging revealed that MET nanoparticles penetrated through the *C. albicans* biofilm matrix and bound to fungal cells. The activity of MET-AmB was no different from the activity of AmB alone against *C. albicans* cells in suspension (MET-AmB MIC90 = 0.125 μg mL^−1^, AmB alone MIC90 = 0.250 μg mL^−1^). MET-AmB eye drops were stable at room temperature for at least 28 days. Conclusions: These biofilm activity findings raise the possibility that MET-loaded nanoparticles may be used to tackle *Candida* biofilm infections, such as refractory ocular fungal infections.

## 1. Introduction 

Biofilms are formed on biotic and abiotic surfaces and are sometimes associated with the flow of fluid across a surface in the presence of the pathogen [1]. Adherence of the microbial cells to the surface is followed by the secretion of extracellular polymer substances, the three-dimensional arrangement of the cells into microcolonies, development of hyphal cells, and ultimately a layer of microbial cells encased in an extracellular polymer matrix [1,2]. Fungal biofilms are composed of fungal cells, the extracellular matrix, and occasionally the presence of hyphal cells [2]. Fungal biofilms develop fully within 38–72 h in vitro, within 24–48 h in vivo, and their establishment is followed by the budding off of cells to develop daughter colonies [3]. Fungal biofilms confer resistance to therapy and result in chronic infections [2,4]. *Candida* species are highly prevalent and well known for their ability to produce opportunistic infections in immunocompromised individuals [4,5]. *Candida* biofilms consist of a complex architecture of an extracellular secreted polysaccharide-protein-lipid-based matrix [6], embedding different budding and filamentous morphological forms of *Candida* cells [7,8]. They are characterised by the expression of adhesion genes [2,9]. The origin of the resistance to therapy lies in a number of biofilm features: the high density of cells within the biofilm, the limited growth of the cells due to a lack of nutrients within the biofilm matrix making them resistant to antifungal drugs, the physical barrier of the matrix itself, the presence of persister cells which are highly resistant to drug therapy, antifungal resistance gene expression, and the increase in cell membrane sterol content [10]. Which factors are indeed dominant in the contribution to resistance are still the subject of debate and here we focus on the physical barrier to the fungal cells, offered by the biofilm itself. 

Biofilms of *C. albicans* and *C. glabrata* were studied here as they are the most common *Candida* species infecting human hosts [3]. These biofilms are very different. *C. glabrata* biofilms are characterized by fungal cells embedded within a protein and carbohydrate matrix, while *C. albicans* biofilms typically contain blastospores and filamentous hyphae all surrounded by a polysaccharide and protein matrix containing water channels [3]. *C. albicans* biofilms are more morphologically heterogenous. 

*Candida* biofilms are usually associated with oral and vaginal candidiasis as well as with implanted medical devices such as vascular and urethral catheters, prosthetic heart valves, and dentures [3]. However biofilms have been implicated in ocular fungal infections, with an ocular biofilm reported in a patient with refractive infectious crystalline keratopathy due to *C. albicans* [11], biofilm forming *C. albicans* species isolated from patients [12], and biofilm forming keratitis patient isolates of *Fusarium solani*, *C. sphaerospermum*, and *Acremonium Implicatum* reported [13]. It is safe to assume that fungal biofilms are a feature of ocular fungal infections, especially those showing drug resistance. Strategies to address ocular fungal biofilms are thus warranted as these biofilm-associated fungal infections are sight-threatening.

Amphotericin B (AmB) is a polyene macrolide isolated from *Streptomyces nodosus* and approved for the systemic treatment of fungal infections [14]. The drug binds to ergosterol, a component of fungal membranes [15]. This binding alters membrane permeability and causes leakage. Depending on drug concentration, AmB has fungistatic or fungicidal activity [14]. AmB is usually used in life threatening infections, although it has also been used for ocular fungal infections as eye drops, subconjunctival, intravitreal, or intravenous injections with varying success [14]. Liposomal AmB and micellar AmB formulations have been shown to have minimal activity in biofilms with minimum inhibitory concentrations 4–8 times higher than seen with planktonic cells [16]. Additionally, AmB’s inherent poor insolubility and subsequent low bioavailability hinders its therapeutic delivery by non-parenteral routes of administration and this low water solubility creates a pharmaceutical formulation challenge. At present, a few liposomal, lipid complex, and colloidal dispersion-based AmB formulations are clinically approved for parenteral administration [17], but their efficacy against *Candida* biofilm infections is modest [16]. AmB encapsulated in N-palmitoyl-N-monomethyl-N,N-dimethyl-N,N,N-trimethyl-6-O-glycolchitosan (Molecular Envelope Technology–MET) nanoparticles is orally active against model systemic *Candida* and *Aspergillus* infections in mice, with nanoparticles being taken up by the gut epithelium [18]. The MET nanoparticles, themselves, are also mucoadhesive [19,20] and are ocular penetration enhancers [21,22]. While much work is ongoing to develop new anti-biofilm agents [23] and anti-biofilm combination therapies [23,24,25], we decided to investigate if the ocular drug penetration enhancer, MET [21,22], could penetrate biofilms with encapsulated AmB and thus show activity against fungal biofilms. We thus set out to evaluate the antibiofilm efficacy of MET-AmB towards *C. albicans* and *C. glabrata* biofilms.

## 2. Materials and Methods

### 2.1. Materials

All materials were obtained from Sigma Aldrich (Gillingham, UK) unless otherwise stated. All solvents were obtained from Fisher Scientific, Loughborough, UK. Amphotericin B (AmB) was obtained from Cayman Chemical (Cambridge Bioscience), Cambridge, UK. *Candida* species were obtained from Thermo Scientific, Loughborough, UK. Menadione was obtained from MP Biomedicals, Irvine, CA, USA. Texas Red succinimidyl ester, Concanavalin A Alexa Flour™ 488 Conjugate, Film Tracer™ SYPRO™ Ruby Biofilm Matrix Stain, and 6-diamidino-2-phenylindole (DAPI) were all obtained from Molecular Probes, Loughborough, UK.

### 2.2. AmB Formulation for Biofilm Studies

MET-AmB nanoparticles were prepared according to a previously published method [18]. Briefly AmB (4 mg) and the MET polymer (Lot No. GCP15Q18, 20 mg) were dissolved in sodium hydroxide (0.02 M, 1 mL, pH = 12) and mixed thoroughly and the pH adjusted to pH = 5 using HCl (0.1 M). The formulation was then centrifuged (13,000 rpm, 30 min, Heraeus Biofuge Fresco, Thermoscientific, Loughborough, UK) to remove any unencapsulated precipitated drug and the supernatant containing the MET-AmB nanoparticles collected. 

Poly(ethylene glycol) (PEG) formulations were produced using a previously published method [26]. This formulation was chosen as a control as it had been used clinically in vaginal candidiasis [26], a disease that has been associated with biofilm formation in mouse models [27] and clinical isolates [28]. Briefly, the PEG suppository base (75% *w*/*w* PEG, Mn = 950–1050 Da, 25% *w*/*w* PEG, Mn = 3350 Da, 1 g) was heated to 65 °C and solubilised in MilliQ water at a 1: 1 volume. AmB was then added to the PEG base aqueous liquid at a concentration of 50 μg mL^−1^ AmB to produce the PEG-AmB formulation. 

MET-AmB nanoparticles were analysed for AmB content using a gradient HPLC method. A standard curve was prepared by diluting AmB in a solvent mixture (acetonitrile, acetic acid, water—52:43:7 volumes) at concentrations ranging from 0.048–50 µg mL^−1^ (peak area = 49.85 [AmB]–55.51, r^2^ = 0.991). MET-AmB formulations were diluted 200-fold using the acetonitrile, acetic acid, water mixture described above and samples (10 µL) chromatographed over a Synergi™ 4-µm Polar-RP 80 Å, LC Column (150 × 4.6 mm, Phenomenex 00F-4336-EQ, Phenomenex, Macclesfield, UK) using an Agilent 1200 Series system (Agilent Technologies, Manchester, UK). Samples were eluted with a gradient mobile phase comprising: A = trifluoroacetic acid (0.1% *v*/*v*), B = acetonitrile: 0 min = A, B (80:20), 6 min = A, B (10:90), 10 min = A, B (80:20). The flow rate was 1.5 mL per minute, the column temperature was 40 °C and the run time was 13 min. Samples were analysed by UV detection at a wavelength of 406 nm. 

MET-AmB particle size was analysed by dynamic light scattering (DLS) using a Malvern Zetasizer Nano series Nano ZS (Malvern Instruments, Malvern, UK). Measurements were carried out at 25 °C, a laser wavelength of 633 nm and a scattering angle of 173°. Undiluted samples were analysed and the data was analysed by the Contin method of analysis. Prior to all measurements, polystyrene standards (diameter = 100 nm) were measured and the size results were in accordance with the manufacturers specifications.

### 2.3. AmB Eye Drops

An alkaline glycerol solution was prepared by adding glycerol solution (1 M, 6 mL) to sodium hydroxide (0.02 M, 14 mL), adjusting the pH to pH = 12 using sodium hydroxide (0.1 M). To this alkaline glycerol solution (0.03 M, pH = 12, 6.5 mL), AmB (13 mg) and the MET polymer (Lot No. GC15/11P21Q15CS, 65 mg) were added. This liquid was mixed and the pH adjusted to pH = 5.4 using hydrochloric acid (1 M). The resulting yellow liquid was filtered (0.22 μm) to simulate sterile filtration and the pH adjusted to pH = 7.0 using sodium hydroxide (1 M). The MET-AmB eye drops were then characterised. 

MET-AmB eye drops were analysed using an isocratic HPLC method. A standard curve was prepared by diluting AmB in the mobile phase at concentrations ranging from 10–125 µg mL^−1^ (peak area = 19.46 [AmB] + 25.96, r^2^ = 0.987). MET-AmB eye drops were diluted 20-fold using the mobile phase and samples (5 µL) chromatographed over a HyperClone™ 5-µm BDS C18 130 Å, LC Column (250 × 4.6 mm, Phenomenex, Macclesfield, UK) using an Agilent 1200 Series system (Agilent Technologies, Manchester, UK). Samples were eluted with a mobile phase comprising acetonitrile, glacial acetic acid, and water (52:4.3:43.7, pH = 2.9). The flow rate was 1 mL per minute, the column temperature was 25 °C and the run time was 12 min. Samples were analysed by UV detection at a wavelength of 406 nm. 

Particle size was analysed as outlined above except that samples were diluted 10 times with MilliQ water prior to analysis and the analysis carried out using disposable polystyrene 70-µL cuvettes. Particle zeta potential was measured as three consecutive measurements using the Zetasizer Nano ZS (Malvern Instruments, Malvern, UK) with the temperature set at 25 °C. The samples were diluted 10 times with MilliQ water and introduced in disposable folded capillary cells. Formulation osmolarity was measured on 100 µL of the undiluted formulation with a Loser Micro-Digital Osmometer Type 5R (Camlab, Cambridge, UK). The pH of the formulation was measured with a PHS-W Series Benchtop pH/mV Meter (SciQuip, Newtown, UK). Stability studies were conducted on the MET-AmB eye drops at 5 °C, room temperature (RT), and at 40 °C. The formulations were characterised for drug content, pH, osmolarity, particle size, and zeta potential at various times during storage. 

### 2.4. AmB Activity in Biofilms 

Biofilms of *C. albicans* ATTC 66027 and *C. glabrata* ATTC 66032 were developed using a previously reported procedure [29]. Briefly, a loop full of *C. albicans*, or *C. glabrata* colonies grown on Sabouraud Dextrose Agar (SDA) plates was inoculated into sterile Yeast Extract Peptone Dextrose (YPD) broth (20 mL) contained in a sterile 50-mL volume conical bottomed tube and incubated overnight (12–16 h at room temperature (16–25 °C)). The following day the cells were harvested by centrifugation (3000× *g* for 5 min at 4 °C, Heraeus Biofuge Stratus Centrifuge, Thermo Fisher Scientific, Germany). The supernatant was discarded and the cells washed with sterile phosphate buffered saline (PBS, pH = 7.4, 2 times 20 mL) with centrifugation (3000× *g* for 5 min at 4 °C) in between to isolate the cells. The fungal cells were resuspended in sterile ice-cold PBS (10 mL) with vigorous vortexing and diluted in a RPMI-1640 medium to obtain 10^6^ cells mL^−1^. The absorbance of the diluted cell suspension was measured at a wavelength of 590 nm and an absorbance value of 0.008 represented a concentration of 10^6^ fungal cells mL^−1^. The volume was adjusted as needed to give a concentration of 10^6^ fungal cells mL^−1^. A 96 well polystyrene, flat bottomed microtiter plates (Nunclon™ Delta surface, Thermo Fisher Scientific, Loughborough, UK) were seeded (10^6^ fungal cells mL^−1^, 100 μL per well). The plates were sealed and incubated at 37 °C, for 24 h. The media was discarded and the cells washed two to three times with ice-cold PBS (200 μL per well). Serial dilutions (0.0025–50 μg mL^−1^) of the different amphotericin B formulations were prepared in a RPMI-1640 cell culture medium on a separate plate and an aliquot (100 μL) was added to each of the wells. The microtitre plate was once again sealed and incubated at 35 ± 2 °C for 24 h. At the end of the incubation period, a menadione–XTT [(2,3-bis-(2-methoxy-4-nitro-5-sulfophenyl)-2H-tetrazolium-5-carboxanilide)] solution (100 μL) containing menadione (1 μM) and XTT (0.74 mM) was added to the wells. The microtitre plates were protected from light and the colour allowed to develop over a 2-h period. The colourless menadione–XTT solution was converted to an orange colour in direct proportion to the number of live fungal cells present. At the end of the incubation period, an aliquot (75 μL) of the supernatant was transferred to a new microtitre plate and the absorbance read (λ = 530 nm) by a plate reader (Spectrostar Omega, BMG Labtech, Aylesbury, Germany). The blank value obtained in the absence of fungal cells was subtracted from all other wells and the EC50 was calculated from the data by assuming the cells in the wells incubated in the absence of AmB represents 100% growth and the wells incubated in the absence of fungal cells represents 0% growth.

### 2.5. AmB Activity in Planktonic C. albicans Cell Suspensions 

The minimum inhibitory concentration (MIC, lowest concentration that inhibits at least 90% growth compared to the drug free control) for MET-AmB and AmB alone were determined using the methodology defined by EUCAST [30]. Briefly, first, an aliquot (100 μL) of the test item (MET-AmB–0.004–2 μg mL^−1^, MET polymer alone–0.014–7μg mL^−1^, or AmB alone–0.02–10 μg mL^−1^) in RPMI-1640 medium containing glucose (2% *w*/*v*) was added to various wells of polysterene, 96-well flat bottomed microtitre plates. To these wells, *C. albicans* (ATTC 66027) cells in RPMI-1640 medium containing glucose 2% *w*/*v* (10^5^ cells mL^−1^) were added. Control wells contained medium with glucose and *C. albicans* cells alone respectively. The plates were sealed, incubated at 37 °C, for 24 h and then read at a wavelength of 530 nm on a plate reader (Synergy HT, Biotek, Santa Clara, CA, USA). The values in the wells devoid of fungal cells was subtracted from all other wells. 

### 2.6. Texas Red (TR) Labelling of MET Polymer 

Texas Red (TR) labelled MET polymer was prepared as previously described [19]. Briefly, the MET polymer (LOT No. GC10P19Q12AA, 100 mg, 11 µmoles) was dissolved in dimethyl sulfoxide (3 mL) and to this was added triethylamine (98 mg, 0.97 mmoles). Texas Red succinimidyl ester (TR-SE, 5 mg, 6 µmoles) was dissolved in DMSO (0.5 mL), added to the MET polymer mixture drop wise with continuous stirring, and the reaction left to proceed for 16 h, protected from light. The mixture was transferred to a 100-mL volume conical flask and precipitated with an ethanol, diethyl ether mixture (1:4, 200 mL). The precipitate was collected and dried. It was then dissolved in methanol (40 mL). To this, water (20 mL) and hydrochloric acid (4 M, 30 μL) were added and the mixture filtered by centrifugation (5000× *g* × 30 min, HERMEL Z 323K, Hermle Labor Technik GmbH, Wehingen, Germany) twice in the presence of Amicon spin filters (3 kDa, 12 mL) to remove any remaining free TR. The retentates from the two filtration steps were combined and further diluted with distilled water (50 mL, to reduce the methanol concentration) prior to dialysis first against sodium bicarbonate solution (0.01 M, 1000 mL) for 5 h with 3 changes. Dialysis was completed by dialysis against distilled water (1000 mL) for 24 h with six changes. The dialysate was freeze-dried, and checked to ensure that it was devoid of any residual free TR using the gel permeation chromatography method described previously [19]. The Texas Red-MET (TR-MET) polymer was reserved for further studies. 

### 2.7. Confocal Laser Scanning Microscopy (CLSM)

*C. albicans* cells in RPMI-1640 (10^6^ fungal cells mL^−1^), prepared, as previously described, were added on to microscopic square glass coverslips, precoated with Collagen type I (0.5 mg mL^−1^), and added to the wells of 24 well polystyrene, flat bottomed tissue culture plates. The cells were allowed to grow at 35 °C, for 7 days and formed a dense biofilm matrix on the coverslip. Exhausted media were exchanged with fresh media daily. TR-MET polymer (10 mg mL^−1^) was added to RPMI-1640 media to form a dispersion of TR-MET nanoparticles (200, 100, and 50 µg mL^−1^) and aliquots (1 mL) dispensed into each of the 24 well plate wells. A negative control was produced by adding RPMI-1640 medium alone to the wells. The wells were incubated at 35 °C for 24 h, media discarded, and wells washed with PBS at 25 °C. The biofilm was successively stained with Concanavalin A Alexa Flour™ 488 Conjugate to stain the cell surface sugars, Film Tracer™ SYPRO™ Ruby Biofilm Matrix Stain to stain the biofilm matrix proteins, and 6-diamidino-2-phenylindole (DAPI) to stain the nucleus, all according to the manufacturer’s instructions. Concanavalin Alexa Flour™ 488 Conjugate in RPMI-1640 medium (25 ug mL^−1^, 0.5 mL) was added to the corresponding well of polystyrene, flat-bottomed 24 well-tissue culture plates, and the wells incubated at 37 °C for 30 min, protected from light. The medium was discarded, replaced, and the whole set up protected from light. For the Film Tracer™ SYPRO™ Ruby Biofilm Matrix staining, medium was discarded from the well, and directly replaced with the Film Tracer™ SYPRO™ Ruby Biofilm Matrix Stain (200 µL) and wells incubated at 25 °C for 30 min, protected from light. Once again, the dye was discarded and replaced with a DAPI nucleic acid stain in the RPMI-1640 medium (20 µg mL^−1^). The wells were incubated at 25 °C, for 5 min, protected from light, and the dye once again discarded. The cells were finally washed with PBS.

In a different set of experiments, *C. albicans* cells, prepared as previously described, were also added onto microscopic square glass uncoated coverslips embedded into wells of polystyrene, flat bottomed 24 well tissue culture plates, and allowed to grow at 35 °C, for 24 h. The cells were then washed with PBS and TR-MET added. Cells dosed with TR-MET were then incubated at 35 °C, protected from light for 24 h, and the cells then washed with PBS (pH = 7.4). Cells (cell membranes) were then stained with CellMask™ Deep Red plasma membrane stain (following a 1000-fold dilution of the manufacturer’s supplied stain). The cells were washed once again with PBS. After the PBS washing, the cells were incubated with the DAPI nucleic acid stain at 25 °C for 5 min and protected from light. The cells were again washed with PBS. Finally, the stained biofilm was covered with a round coverslip, 1 drop of antifade added, and the coverslip sealed in place using nail polish. Slides were imaged using an oil objective on a confocal laser scanning microscope (Carl Zeiss LSM 710, Zeiss, Oberkochen, Germany). Image capturing and processing was carried out using the ZenPro software (Zeiss, Oberkochen, Germany).

## 3. Results

### 3.1. Formulations

The MET-AmB nanoparticle eye drop formulation was stable for up to 4 weeks at refrigeration temperature in terms of drug content and although there were slight changes in the level of AmB seen at intermediate time points, the drug content measured at 4 weeks showed exceptional stability at all storage conditions (Table 1). A 20–50% increase in particle size was noted with the formulations, but the formulation remained in the colloidal size range (<500 nm) for 4 weeks under all storage conditions. There were slight changes in pH and the pH of the formulation ranged from 6.8 to 7.2 over the 4-week period, but still remained within the ocular comfort range of pH = 6.8–7.9 [31]. Overall, the formulation showed good stability over the 4-week period, irrespective of the storage conditions.

The MET-AmB nanoparticles formulation for biofilm studies were prepared with the following characteristics: Drug content = 3.08 ± 0.53 mg mL^−1^, particle size diameter = 157 ± 3 nm, and PDI = 0.43 ± 0.06. Figure 1 shows that the MET-AmB nanoparticles showed superior activity against both *C. albicans* and *C. glabrata* biofilms when compared to AmB alone and when compared to a PEG-AmB formulation that has previously been used clinically for the treatment of non-albicans vaginal candidiasis [26]. However, activity against *C. Albicans* cells in suspension was similar when AmB alone was compared to the MET-AmB nanoparticle formulation (Table 2). The MIC values are similar to those reported by others for susceptible *C. albicans* strains (<1 μg mL^−1^) [32]. This provides evidence of the increased efficacy of nanoparticle formulation within the biofilm.

### 3.2. Confocal Laser Scanning Microscopy

Having observed the superior activity of MET-AmB in killing fungal cells in biofilms, we then sought to test the hypothesis that the MET enables superior penetration of encapsulated AmB into the biofilm. MET is a mucoadhesive polymer, which penetrates gut mucus [19] and shows a long residence time on mucosal epithelia [33]. Using live cell imaging on non-fixed cells, we established that MET nanoparticles do indeed penetrate the biofilm and adhere to the fungal cell walls (Figure 2b,d and Figure 3).

## 4. Discussion

While it must be acknowledged that the biofilm features that confer drug resistance are multifactorial, as described above, with no specific feature known to critically drive drug resistance, our data show that MET nanoparticles penetrate within *C. albicans* biofilms (Figure 2 and Figure 3) in vitro and MET-AmB formulations show superior anti-fungal activity against biofilm-embedded *Candida* fungal cells (Figure 1). Our data suggest that materials that penetrate biofilms may be used to tackle biofilm-related resistance and it is possible that the physical barrier posed by the biofilm is the main driver of resistance. As biofilms are implicated in ocular fungal diseases [11], especially resistant infections, this finding means that MET-AmB eye drop formulations may provide a superior therapeutic outcome in biofilm-associated ocular fungal infections. The issue of penetration into the biofilm being a driver of resistance is controversial in the literature as reduced penetration of AmB into *C. albicans* biofilms has been implicated in drug resistance [34], although others have reported that penetration of AmB into *C. albicans* biofilms has been shown to be adequate (concentrations in excess of the MIC of *C. albicans)* and yet still not achieve fungal cell kill within the biofilm [35]. Our data suggest that MET nanoparticles achieving good penetration within the biofilm to allow close proximity of the encapsulated drug to the fungal cells could be the cause of the moderation in drug resistance seen. The MET-AmB formulation is not superior to AmB alone when fungal *C. albicans* cells are in suspension and the MET itself is not active at low concentrations against *C. albicans*, indicating that the MET has no intrinsic fungicidal properties at the concentrations used (Figure 2, Table 2). Our data suggest that the MET particles appear to act by being able to deliver AmB to fungal cells buried within the biofilm matrix. The mechanism by which MET nanoparticles penetrate the biofilm matrix is unclear at present. MET nanoparticles are positively charged at neutral pH [22] and MET-AmB formulations are also positively charged at neutral pH (Table 1) and this positive charge may promote an association with the biofilm matrix. The AmB resistance seen with *C. albicans* biofilms has been attributed to AmB binding to 1,3-glucans of the biofilm matrix [36]. It is feasible to assume that encapsulated AmB would be less likely to bind to these biofilm matrix molecules and it is possible that both of these processes (penetration of encapsulated drug into biofilm and the drug not binding specifically to the biofilm matrix) may be operational in our work. We further speculate that once the MET nanoparticles are bound to the cell wall, AmB would be released from these bound nanoparticles and be able to lyse the fungal cells. 

The differential activity of MET-AmB, when compared to AmB alone, that is seen against *C. glabrata* biofilms is not as pronounced as that seen with *C. albicans* biofilms (Figure 2). *C. glabrata* develops thinner biofilms (~25 µm) compared to thicker ones (~100 µm) of *C. albicans* in RPMI culture media [9]. Furthermore, *C. glabrata* biofilms lack the filamentous entangling cell forms seen in *C. albicans* biofilms [37]. C. albicans biofilms are more morphologically heterogenous and consist of oval budding yeast species, hyphae and pseudohyphae, all contained within a matrix of proteins, carbohydrates (β-1,6-glucans and mannose), and water channels [3]. Incidentally, β 1,6-glucans and mannose are critical compositions of the *C. albicans* biofilm, as deletion of the genes encoding for the proteins that produce these sugars results in defective biofilms [38]. The difference in thickness and structure of biofilm matrices between the two *Candida* species might explain the reduced decrease in EC50 values seen with MET-AmB formulations in *C. glabrata* biofilms (Figure 2). For example, the presence of the water channels in *C. albicans* biofilms may aid particle diffusion within the biofilm.

## 5. Conclusions

We demonstrated that a MET-AmB formulation is a more active antifungal formulation when compared to the drug alone in *Candida* biofilms and that MET nanoparticles penetrate into the biofilm and adhere to the fungal cell walls. The MET-AmB eye drop formulation also developed during this work may serve as a method to tackle ocular fungal biofilms. 

## Figures and Tables

**Figure 1 pathogens-11-00073-f001:**
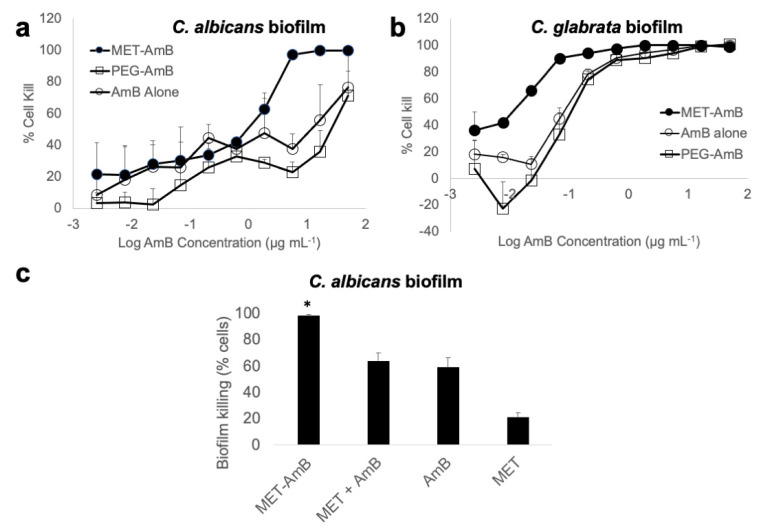
Antifungal activity of AmB formulations in *Candida* biofilms grown at 35 °C, for 24 h (RPMI-1640 media) in polystyrene flat bottomed 96-well microtiter plates and the average of three separate experiments: (**a**) Anti-fungal activity in *C. albicans* biofilms (EC50 MET-AmB = 1.18 ± 0.54 μg mL^−1^, EC50 PEG-AmB = 29.09 ± 12.18 μg mL^−1^, EC50 AmB alone = 24.74 ± 22.72 μg mL^−1^, EC50 MET alone—data not shown - > 300 μg mL^−1^), (**b**) anti-fungal activity in *C. glabrata* biofilms (EC50 MET-AmB = 0.03 ± 0.01 μg mL^−1^, EC50 PEG-AmB = 0.29 ± 0.06 μg mL^−1^, EC50 AmB = 0.08 ± 0.03 μg mL^−1^), and (**c**) antifungal activity in *C. albicans* biofilms with various AmB formulations containing 6 μg mL^−1^ AmB (MET-AmB = AmB encapsulated within MET nanoparticles, MET + AmB = empty MET nanoparticles added with AmB solution, AmB = AmB alone, MET = MET alone devoid of AmB), * = statistically significantly different from all other formulations (*p* < 0.05).

**Figure 2 pathogens-11-00073-f002:**
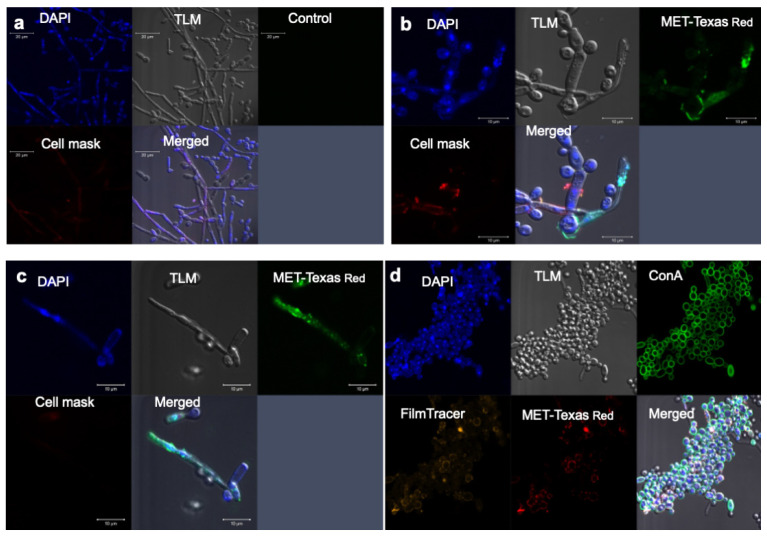
Confocal laser scanning microscopy cell imaging of *C. albicans* biofilms grown in RPMI-1640 media at 35 °C—(**a**) biofilm grown on a microscope cover slip for 24 h with no treatment, (**b**) biofilm grown on a microscope cover slip for 24 h and then treated with MET-Texas Red (50 μg mL^−1^) for 24 h, (**c**) biofilm grown on a microscope cover slip for 24 h and then treated with MET-Texas Red (100 μg mL^−1^) for 24 h, and (**d**) biofilm grown on a collagen-coated microscope cover slip for 7 days and then treated with MET-Texas Red (50 μg mL^−1^) for 24 h.

**Figure 3 pathogens-11-00073-f003:**
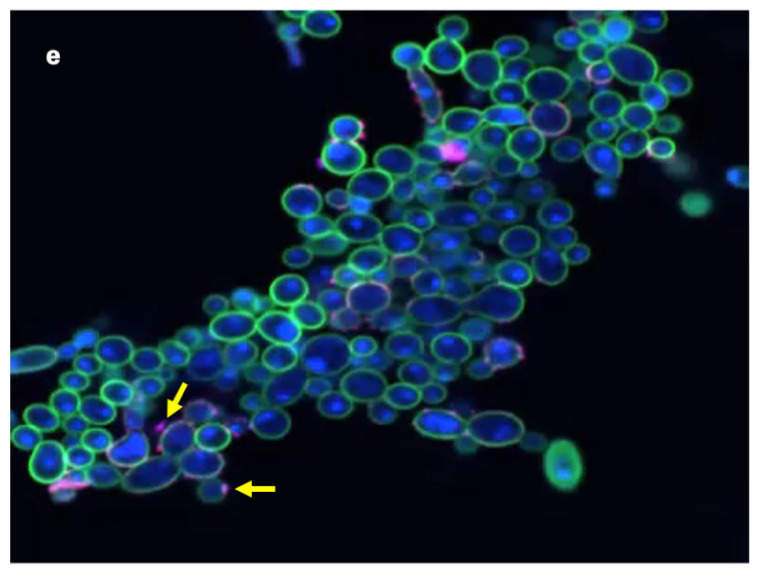
Zoomed in merged image from Figure 2d, showing MET-Texas Red within the biofilm and bound to *C. albicans* fungal cells (arrowed). Appendix A.

**Table 1 pathogens-11-00073-t001:** Characteristics of the MET-AmB (Molecular Envelope Technology—Amphotericin B) eye drops stored at different temperature conditions over 28 days presented as mean ± standard deviation, (n = 3).

Time Point		Concentration (mg mL^−1^)	(%) Remaining Drug Content	z-Average (nm)	Polydispersity Index (PDI)	Zeta Potential (mV)	pH	Osmolarity (mOsm)
Day 0		1.81 ± 0.06	-	59.7 ± 1.4	0.381 ± 0.050	28.0 ± 0.6	7.2 ± 0.1	355 ± 19
Week 1	5 °C	1.90 ± 0.06	106.7 ± 4.2	77.9 ± 5.9 *	0.252 ± 0.020 *	27.4 ± 0.9	6.9 ± 0.0 *	331 ± 3
	RT	1.81 ± 0.07	101.4 ± 3.3	74.4 ± 5.2 *	0.241 ± 0.019 *	27.3 ± 0.6	6.8 ± 0.1	339 ± 7
	40 °C	1.86 ± 0.04	104.1 ± 1.9	88.1 ± 2.2 *	0.231 ± 0.018 *	28.7 ± 1.0	6.8 ± 0.0 *	330 ± 4
Week 2	5 °C	1.84 ± 0.07	103.4 ± 3.7	79.6 ± 5.2 *	0.245 ± 0.018 *	26.0 ± 0.9	7.0 ± 0.0 *	334 ± 4
	RT	1.74 ± 0.03*	97.8 ± 3.1	76.2 ± 4.4 *	0.235 ± 0.012 *	27.0 ± 0.8	7.1 ± 0.0	335 ± 5
	40 °C	1.72 ± 0.03	96.5 ± 2.3	89.4 ± 2.8 *	0.225 ± 0.010 *	28.2 ± 3.4	7.0 ± 0.0 *	336 ± 4
Week 4	5 °C	1.91 ± 0.03	107.1 ± 1.9	85.9 ± 6.7 *	0.234 ± 0.010 *	28.3 ± 0.7	7.1 ± 0.0	338 ± 4
	RT	1.89 ± 0.04	106.0 ± 1.0	80.2 ± 4.7 *	0.225 ± 0.006 *	26.7 ± 1.3	7.1 ± 0.0	334 ± 4
	40 °C	1.81 ± 0.11	100.1 ± 4.2	89.5 ±1.1 *	0.229 ± 0.008 *	29.7 ± 1.0	6.9 ± 0.1 *	336 ± 3

* significantly different to the formulation on day 0 (*p* < 0.05).

**Table 2 pathogens-11-00073-t002:** MIC of formulations against *C. albicans* planktonic cells in suspension.

Formulation	MIC (μg mL^−1^)
MET-AmB	0.125
AmB	0.25
MET	>7

## Data Availability

The data presented in this study are available on request from the corresponding author. The data are not publicly available due to commercial confidentiality reasons.

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
