# Peer review of "Amphotericin B Polymer Nanoparticles Show Efficacy against Candida Species Biofilms"

_pathogens, 2022, doi:10.3390/pathogens11010073_

Round 1

Reviewer 1 Report

The manuscript is interesting and the obtained research results are valuable. Amphotericin in the form of nanoparticles is an interesting proposition in the form of a drug for combating Candida infections. The results suggest that substances such as MET-AmB in the form of nanoparticles penetrate the structure of the Candida biofilm may serve as a method to tackle ocular fungal biofilms.

No information about the applied Candida standard strain for the determination of MIC values; new EUCAST recommendations: EUCAST DEFINITIVE DOCUMENT E. DEF 7.3: EUCAST methods for antimicrobial susceptibility testing of yeasts, moulds and dermatophytes; (cited document from 2008).

Author Response

REVIEWER

No information about the applied Candida standard strain for the determination of MIC values; new EUCAST recommendations: EUCAST DEFINITIVE DOCUMENT E. DEF 7.3: EUCAST methods for antimicrobial susceptibility testing of yeasts, moulds and dermatophytes; (cited document from 2008).

AUTHOR RESPONSE

All changes to the manuscript appear in red font.

We have included the ATTC reference details in Line 207. We have also provided a reference to other MIC data in the literature to show the comparability between our data and data in the literature and this appears in Lines 302-303.

We have also edited the manuscript to clarify various points and these appear in red font in Lines 35-43 and Lines 355-368.

Reviewer 2 Report

The manuscript entitled “Amphotericin B - polymer nanoparticles show efficacy against Candida species biofilms” describes very interesting scientific issue. The results provide an interesting added value in our comprehension how reformulation of well-known antifungal drugs like Amphotericin B may improve its efficacy, and justify of the publication of this manuscript in the Pathogens. Although some fragments were difficult to read and several inaccuracies in the methods and results sections were rather detracting.

Some corrections must be introduced:

  1. The introduction section should be enriched with information on the purposefulness of selecting strains: albicans and C. glabrata. Although biofilm formation is true for every Candida species, there are differences between C. albicans and C. glabrata biofilms, regarding their dimensions and structure, cell morphology, EPS (extracellular polymeric substances) produced and secreted, response to environmental cues, and resistance to antifungal drugs. Please, provide also a more in-depth analysis of the differences in MET-AmB nanoparticle eye drop formulation antibiofilm activity in this context in the Discussion section.
  2. Figure1c – what does “MET + AmB” mean? It is not clear and is not explained in the figure description. Shouldn't it be “PEG-AmB”?
  3. Line272-280 should be stated just after Table 1.
  4. What does “EC90” mean in table 2? I think that it is MIC90 value presented.
  5. Figure 2e should be presented as separate Figure 3.
  6. It is known that C. albicans cells incubated in RPMI-1640 for 7 days at 35°C will form a dense biofilm matrix although surprisingly, no mycelial forms of C. albicans were observed in the microscopic image (Figure 2d,e). Please explain why?

Author Response

REVIEWER

The introduction section should be enriched with information on the purposefulness of selecting strains: albicans and C. glabrata. Although biofilm formation is true for every Candida species, there are differences between C. albicans and C. glabrata biofilms, regarding their dimensions and structure, cell morphology, EPS (extracellular polymeric substances) produced and secreted, response to environmental cues, and resistance to antifungal drugs.

AUTHOR RESPONSE

We have added new information to the introduction to further describe the differences between the two Candida strains. This appears in Lines 57-62.  We have added additional information on biofilm formation in Lines 36-43.  We are focused on amphotericin B formulations and biofilms and so have limited our references to amphotericin B activity.  Lines 301-302 provide data from the literature on the MIC of amphotericin B in susceptible Candida strains.

REVIEWER

Please, provide also a more in-depth analysis of the differences in MET-AmB nanoparticle eye drop formulation antibiofilm activity in this context in the Discussion section.

AUTHOR RESPONSE

We have added more analysis of the results in the context of the biofilm structure and biochemistry and these appear in Lines 323 - 327 and Lines 355 - 368.

REVEIWER

Figure1c – what does “MET + AmB” mean? It is not clear and is not explained in the figure description. Shouldn't it be “PEG-AmB”?

AUTHOR RESPONSE

This has been clarified in the figure legend in Lines 292-294

REVIEWER

Line272-280 should be stated just after Table 1.

AUTHOR RESPONSE

This paragraph has been moved and now appears in Lines 275 - 283 (just after Table 1.)

REVIEWER

What does “EC90” mean in table 2? I think that it is MIC90 value presented.

AUTHOR RESPONSE

This has been changed to MIC and a definition of MIC moved closer to the first presentation of the initials.  This all appears on Lines 201 - 202.

REVEIWER

Figure 2e should be presented as separate Figure 3.

AUTHOR RESPONSE

This has been relabelled accordingly.

REVIEWER

It is known that C. albicans cells incubated in RPMI-1640 for 7 days at 35°C will form a dense biofilm matrix although surprisingly, no mycelial forms of C. albicans were observed in the microscopic image (Figure 2d,e). Please explain why?

AUTHOR RESPONSE

In essence we were looking to show the nanoparticles bound to the cells and so focused on these areas for imaging.  The hyphae are usually better seen with the scanning electron microscopy images, as shown in the references given below (Eukaryotic Cell 2005, 633–638, Current Biology 2005, R453-R455).  The Supplementary Information video file does show filamentous species as one zooms through the z-axis.  The extracellular polymeric substance matrix is not very visible after sample preparation as outlined in the article (EUKARYOTIC CELL, Apr. 2005, p. 633–638).

 EUKARYOTIC CELL, Apr. 2005, p. 633–638

Current Biology 2005, 15, R453-R455
